# Genome-Wide Characterization of *MADS-box* Genes Identifies Candidates Associated with Flower and Fruit Development in Loquat (*Eriobotrya japonica* Lindl.)

**Wenxiang Li** [1,2,†]**, Xiaopei Liu** [1,2,†]**, Chongbin Zhao** [1,2]**, Wendong Wu** [1,2]**, Yuanyuan Jiang** [3]**, Wenbing Su** [4]**, Shunquan Lin** [1,2]**, Xianghui Yang** [1,2,*]**and Ze Peng** [1,2,*]

1   State Key Laboratory for Conservation and Utilization of Subtropical Agro-Bioresources, South China Agricultural University, Guangzhou 510642, China; wxli@stu.scau.edu.cn (W.L.); luixp_we@163.com (X.L.); zhaocb@stu.scau.edu.cn (C.Z.); wwd13320190594@163.com (W.W.); loquat2021@163.com (S.L.)
2   Key Laboratory of Innovation and Utilization of Horticultural Crop Resources in South China, Ministry of Agriculture and Rural Affairs, College of Horticulture, South China Agricultural University, Guangzhou 510642, China
3   College of Biology and Agriculture, Shaoguan University, Shaoguan 512005, China; yyjiang613@163.com
4   Fruit Research Institute, Fujian Academy of Agricultural Sciences, Fuzhou 350013, China; suwenbing13@163.com
*   Correspondence: zepeng@scau.edu.cn (Z.P.); gzyxh@scau.edu.cn (X.Y.)
†   These authors contributed equally to this work.

**Abstract:** The *MADS-box* transcription factors have garnered substantial attention due to their crucial involvement in various biological processes, particularly in flower organogenesis. A comprehensive investigation into the *MADS-box* genes remains lacking in loquat (*Eriobotrya japonica* Lindl.). In the current study, to preliminarily explore the potential candidate genes related to flower and fruit development, a genome-wide analysis was carried out to identify and characterize the *MADS-box* gene family in loquat. Among the 125 identified *EjMADS-box* members, 49 genes belonged to type I, which were subsequently assigned to three subfamilies: Mα (25 genes), Mβ (10 genes), and Mγ (14 genes). Additionally, 76 genes fell under type II, which were categorized into two groups: MIKC^C (70 genes) and MIKC* (6 genes). Through the collinearity analysis and comparison of the gene numbers between loquat and other Rosaceae genomes, it was revealed that the type II *MADS-box* members were expanded in Maloideae after a whole genome duplication. The gene expression analysis utilizing various tissues during flower development revealed that the expression patterns of the ABCDE model homologs were conserved in loquat. In addition, several candidate genes potentially involved in flower bud differentiation (*EjMADS107/109*) and fruit expansion (*EjMADS24/46/49/55/61/67/77/86*) were identified. This analysis could serve as a fundamental basis for investigating the molecular functions of the *MADS-box* genes in the development of flowers as well as fruits in loquat.

**Keywords:** ABCDE model; floral transition; fruit size; loquat (*Eriobotrya japonica* Lindl.); *MADS-box*

## 1. Introduction

Loquat (*Eriobotrya japonica* Lindl.) is among the most popular fruits in China during the spring season. In comparison to other fruit trees in Rosaceae, loquat exhibits a unique flowering time characterized by the initiation of flower bud formation in summer and blooming in fall, without undergoing dormancy in winter or during specific cold periods in China [1]. In horticultural plants, the initiation and progression of flowers play a crucial role in the successful fruit set. Flower development in plants is a highly complex and ordered process that requires a complex regulatory network to ensure it is in the right environmental conditions for flowering. Within this intricate network, transcription factors hold pivotal significance, actively participating in the orchestration of flower development [2,3]. The MADS-box proteins constitute a significant category of transcriptional regulators that





participate in the flowering process. They assume crucial positions in regulating the intricate processes of floral organ differentiation and formation, as well as pollen development and maturation [4].

Currently, there is a relatively comprehensive understanding regarding the functions of the *MADS-box* genes, with the type II members being of particular interest in plants. For instance, the participation of $MIKC^C$ genes in somatic embryo development has been demonstrated in *Arabidopsis* and soybean [5]. Moreover, these $MIKC^C$ genes are crucial in regulating flower bud differentiation and controlling the timing of flowering. In the model plant *Arabidopsis*, *SUPPRESSOR OF OVEREXPRESSION OF CONSTANS 1* (*SOC1*) has the ability to integrate various signals that facilitate flowering, thereby promoting the process of flower organogenesis [6,7]. The phenomenon of delayed flowering was observed upon the overexpression of *SHORT VEGETATIVE PHASE* (*SVP*), while early flowering was observed in *svp* mutants [8]. *APETALA1* (*AP1*), a floral meristem identity gene, was found to have the ability to trigger flower formation [9].

Research on the $MIKC^C$ genes showed that they were also involved in the organogenesis of plant flowers. The ABC model was previously introduced in the scientific literature [10]. The genes for floral organ characteristics were classified into three classes, including the A-class types such as *AP1* and *AP2*, the B-class types such as *PI* and *AP3*, and the C-class types such as *AG*, which are *MADS-box* genes, except for *APETALA2* (*AP2*) [10,11]. With the advancement of research on flower organ development, the ABC model has been upgraded to a new model named ABCDE. This expansion incorporated additional genes into the model, among which the D-class genes, such as *STK*, are responsible for determining the identity of ovules, while the E-class genes, including *SEP1-4*, synergistically regulate floral organs production [12].

Currently, a few studies investigating the molecular functions of the *MADS-box* genes are available in loquat. Two homologs of *SOC1* were identified in loquat, and they controlled the flowering time [13]. *EjSVP-1* and *EjSVP-2* were functionally different compared to *Arabidopsis*, and the overexpression of the *EjSVP-2* gene did not delay flowering in *Arabidopsis*, but affected the formation of floral organs [14]. The occurrence of double flowering in loquat was observed, with *EjPI* and *EjAG* playing key roles in these two double flowering phenomena, respectively [15,16]. In addition, in loquat, *EjAGL17*, *EjAP1*, and *EjCAL* were implicated in flower bud differentiation [17–19].

Given the importance of the *MADS-box* genes in various developmental processes of plants, especially in flowering and fruit development, it is necessary to fully identify the *MADS-box* genes in the loquat (*Eriobotrya japonica* Lindl.) genome at a genome-wide level. In the current study, 125 *EjMADSs* were discovered in the loquat genome, and we analyzed the characteristics, structures, phylogenetic relationships, and classification of these *EjMADSs*. Furthermore, through the analysis of transcriptome data and qRT-PCR experiments, several potential *EjMADSs* were identified, which were presumed to be significant in the processes of different tissues. Additionally, the investigation successfully identified potential ABCDE model genes for loquat flowers. The findings from this study offer a valuable point of reference for the functional identification and characterization of *EjMADSs*, thereby providing candidate genes for the study of loquat flower and fruit development.

## 2. Materials and Methods

### 2.1. Plant Materials

All plant materials were maintained in the loquat (*Eriobotrya japonica* Lindl.) Germplasm Resource Preservation Garden (South China Agricultural University, Guangzhou, China). The loquat trees were 13 years old and grown under regular management conditions. Using the cultivar 'Jiefangzhong' as the material, five key stages during the flowering process were selected by paraffin section and observation. The methods of paraffin section is referred to in the published literature [13]. Tissues including pistils, stamens, sepals, and petals were collected during the early flowering stage, specifically when approximately

25% of the flowers had bloomed. Subsequently, the collected samples were rapidly frozen using liquid nitrogen and stored in a −80 °C freezer.

### 2.2. Genome-Wide Identification

AtMADSs protein sequences of *Arabidopsis* and VvMADS protein sequences of grape were derived from a previous report [20,21]. To conduct a search for *EjMADSs*, the reference genome of 'Seventh Star' was utilized [22]. The protein sequences of AtMADSs were compared with the protein sequences of loquat using Blastp (e-value $< 1 \times 10^{-5}$, identity > 40%). On the other hand, the Hidden Markov Model (HMM) profile of SRF (type I) domain (PF00319) was downloaded from the Pfam database (https://www.ebi.ac.uk/interpro/, accessed on 6 January 2023), which was used to conduct another search with HMMER v3.3.1 (Cambridge, MA, USA) (e-value $< 1 \times 10^{-5}$) [23]. The non-redundant protein sequences were submitted to the SMART database (https://smart.embl.de/, accessed on 8 January 2023) for confirmation. The identification strategy for *MADS-box* genes in wild loquat (*Eriobotrya japonica*), strawberry (*Fragaria vesca*), apple (*Malus domestica*), pear (*Pyrus bretschneideri*), sweet cherry (*Prunus avium*), *Gillenia trifoliata*, *Prunus mume*, pear (*Pyrus communis*), *Rosa chinensis*, peach (*Prunus persica*), and raspberry (*Rubus occidentalis*) are the same as that for loquat. The genome data of *M. domestica*, *P. mume*, *P. communis*, *P. persica*, *P. avium*, *R. chinensis*, *F. vesca*, and *R. occidentalis* were obtained from the GDR database (https://www.rosaceae.org/, accessed on 1 January 2023); the genome data of *P. bretschneideri* and *G. trifoliata* were downloaded from NCBI database (https://www.ncbi.nlm.nih.gov/, accessed on 1 January 2023); the wild loquat (*E. japonica*) genome data were obtained from the Genome Warehouse (https://ngdc.cncb.ac.cn/gwh/, accessed on 3 January 2023) [24].

### 2.3. Phylogenic Analysis

The protein sequences were aligned utilizing the MAFFT v7.310 software (Osaka University, Osaka, Japan) [25]. TrimAl v1.4 software (CRG, Barcelona, Spain) was employed to eliminate any gaps in the alignment [26]. IQ-TREE v2.2.2.6 (Australian National University, Canberra, Australia) and the maximum likelihood (ML) approaches were utilized for constructing the phylogenetic tree, with 5000 bootstraps [27]. The phylogenetic tree representing the evolutionary relationships among 14 species was constructed using Orthofinder2 v2.5.4 (University of Oxford, Oxford, UK) [28].

### 2.4. Conserved Motif, Functional Domains, and Gene Structure Analysis

Conserved domains were detected by employing the SMART database (https://smart.embl.de/, accessed on 8 January 2023). The gff3 files were utilized for visualizing gene structures using Tbtools v2.012 (South China Agricultural University, Guangzhou, China), a software tool for visualizing genomic data [29]. Conserved motifs were detected by employing the MEME v5.5.4 (https://meme-suite.org/meme/, accessed on 8 January 2023).

### 2.5. Synteny Analysis and Chromosome Location

Synteny analysis was carried out using MCScanX (University of Georgia, GA, USA), a well-established tool for analyzing and comparing gene order and arrangement within genomes. The obtained results were subsequently visualized employing Tbtools [30]. The nonsynonymous/synonymous ratio (Ka/Ks) analysis for the genes linked to duplication events was performed using Tbtools.

### 2.6. RNA-seq Analysis

RNA-seq data were acquired from previously published reports [31,32]. Clean data were obtained using fastp v0.23.2 (HaploX Biotechnology, Shenzhen, China) [33]. HISAT2 v2.2.1 (University of Texas, TX, USA) was employed for aligning the polished RNA-seq reads against the genome of 'Seventh Star' [34]. The mapped reads were subsequently quantified by featureCounts V2.0.6 (The Walter and Eliza Hall Institute of Medical Research,

VIC, Australia) [35]. The transcripts were normalized to TPM values using R, and Tbtools was used for heatmap construction.

### 2.7. RNA Extraction and qRT-PCR

The EASYspin Plus plant RNA extraction kit (Aidlab, Beijing, China) was used for RNA isolation. The PrimeScript™ RT reagent kit (TaKaRa, Kusatsu, Japan) was used for the first-strand cDNA synthesis. BatchPrimer3 was used for qRT-PCR primer design (primer sequences provided in Table S1) [36]. qRT-PCR was carried out following the methodology described in our previous report [37]. *EjACT* was used as the reference gene [38]. The relative expression levels were determined using the $2^{-\Delta\Delta Ct}$ method [39].

### 3. Results

#### 3.1. MADS-box Genes in 12 Rosaceae Species

Following the application of consistent identification criteria, 1062 *MADS*s were identified from 12 Rosaceae species in this study (Figure 1, Table S2). With the exception of *Pyrus communis,* it was observed that species within the apple subfamily exhibited a higher number of identified *MADS-box* genes compared to other Rosaceae species, probably because Maloideae has undergone a recent WGD event (Figure 1) [32]. Loquat (*Eriobotrya japonica*) exhibited the highest number of *MADS-box* members among all the species in this study, with 125 *EjMADSs* in cultivated loquat and 130 in wild loquat.

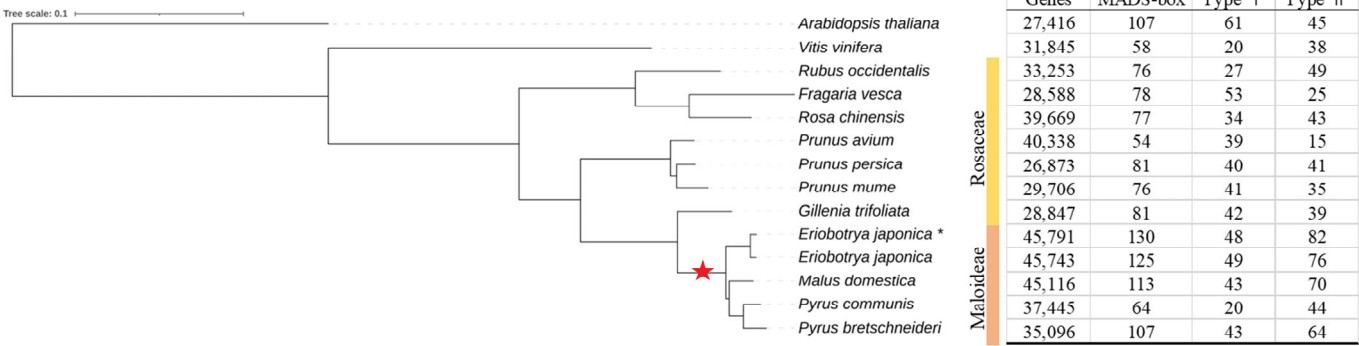

| Genes | MADS-box | Type I | Type II |
|---|---|---|---|
| Arabidopsis thaliana | 27,416 | 107 | 61 | 45 |
| Vitis vinifera | 31,845 | 58 | 20 | 38 |
| Rubus occidentalis | 33,253 | 76 | 27 | 49 |
| Fragaria vesca | 28,588 | 78 | 53 | 25 |
| Rosa chinensis | 39,669 | 77 | 34 | 43 |
| Prunus avium | 40,338 | 54 | 39 | 15 |
| Prunus persica | 26,873 | 81 | 40 | 41 |
| Prunus mume | 29,706 | 76 | 41 | 35 |
| Gillenia trifoliata | 28,847 | 81 | 42 | 39 |
| Eriobotrya japonica * | 45,791 | 130 | 48 | 82 |
| Eriobotrya japonica | 45,743 | 125 | 49 | 76 |
| Malus domestica | 45,116 | 113 | 43 | 70 |
| Pyrus communis | 37,445 | 64 | 20 | 44 |
| Pyrus bretschneideri | 35,096 | 107 | 43 | 64 |

**Figure 1.** The species tree and *MADS-box* gene numbers for fourteen genomes. The phylogenetic tree representing the evolutionary relationships among 14 species was constructed using Orthfinder2 with ML method [28]. The wild loquat was marked with '*'. The red star represents the whole genome duplication (WGD) event.

#### 3.2. Phylogenetic Analysis and Gene Characterization

In order to categorize the 1062 *MADS-box* genes into distinct types, the 105 genes from *Arabidopsis* were used to assist in the classification and construction of phylogenetic trees for each Rosaceae species separately (Figure S1). Subsequently, the type II *MADS-box* members from *Arabidopsis*, rice, as well as grape were utilized to further distinguish subfamilies from all type II *MADS-box* members. The phylogenetic analysis classified the type II *MADS-box* genes into 15 subfamilies (Figure 2). Except for *Pyrus communis* and *Rubus occidentalis*, there was no significant variation observed in the gene numbers of the type I members among the remaining nine Rosaceae species; however, *Eriobotrya japonica*, *Malus domestica*, and *Pyrus bretschneideri* exhibited significantly higher gene numbers of type II members compared with other Rosaceae species, suggesting that these species experienced a significant proliferation of type II genes following the WGD event. On the other hand, the AGL15 subfamily genes were expanded in *Rubus occidentalis*, *Fragaria vesca*, and *Rosa chinensis*, although no recent WGD event has occurred (Figure 2).

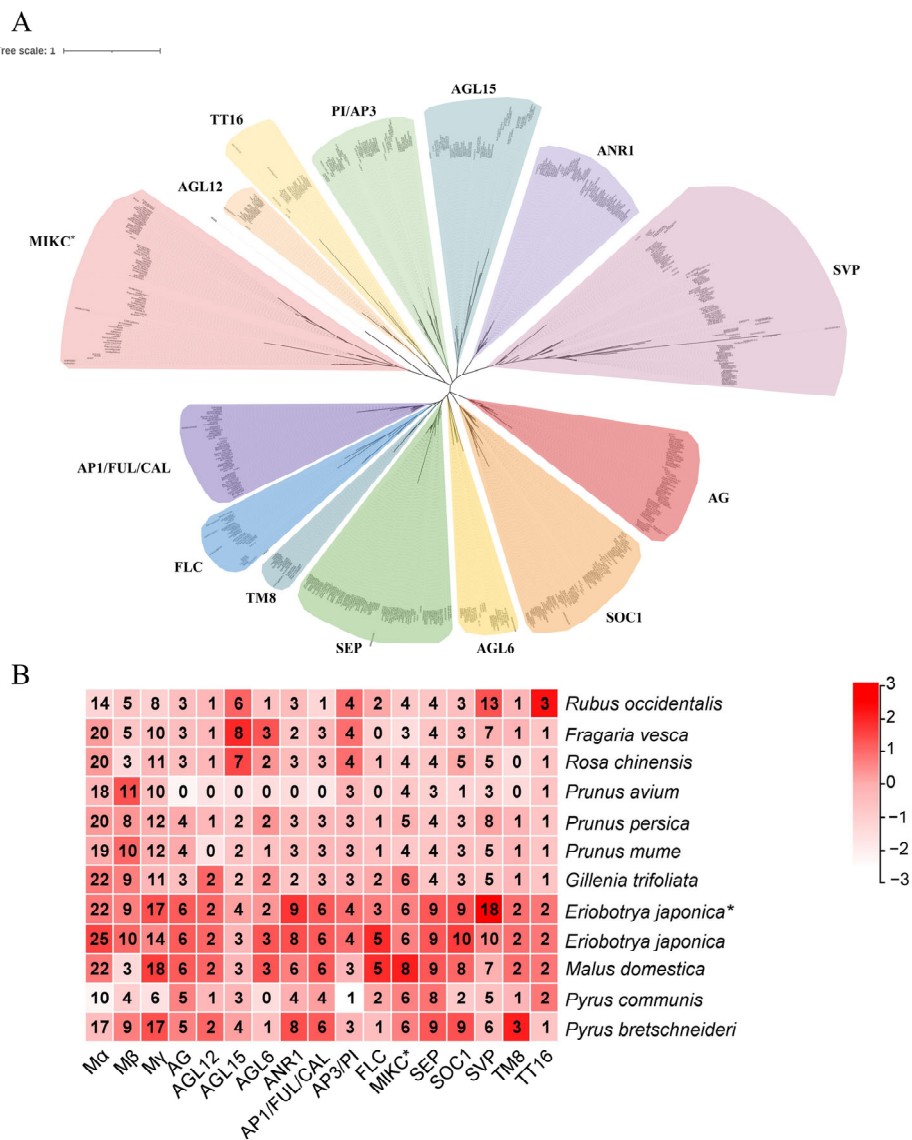

**Figure 2.** Phylogenetic relationships and number of individual subfamily members of MADS-box proteins of 12 Rosaceae genomes. (**A**) The tree was obtained following the execution of multiple sequence alignments employing the MAFFT software with the ML method. Distinctive colors were assigned to 15 clades in order to facilitate visual differentiation. (**B**) The heat map illustrates the distribution of $MIKC^C$ genes across various clades in the Rosaceae species. The wild loquat was marked with '\*'.

### 3.3. Chromosomal Locations and Annotation

In total, 125 *EjMADSs* were identified from the genome of 'Seventh Star'. The nomenclature assigned to these *EjMADSs* was derived from their respective genomic locations, ranging from Chromosome 1 to 17 in a top-to-bottom order (Figure 3). Among the 17 chromosomes of the loquat, Chr2 exhibited the highest abundance of *EjMADS* genes, encompassing a total of 19 genes. In contrast, chromosome 12 had only one *EjMADS81* (Figure 3). The type I genes were found to be dispersed among 13 chromosomes, with the highest gene count observed on Chr 3 with seven genes, with Mα, Mβ, and Mγ were all include (Figure 3). The analysis of the peptide length revealed that the length of the loquat MADS-box proteins varied considerably, with 96 proteins ranging from 200 aa to 400 aa (Table S3). The molecular weight (MW) of the *MADS-box* family members in loquat ranged from 10,332.79 to 73,635.28 Da; the isoelectric points (PI) ranged between 4.07 and 10.21 (Table S3).

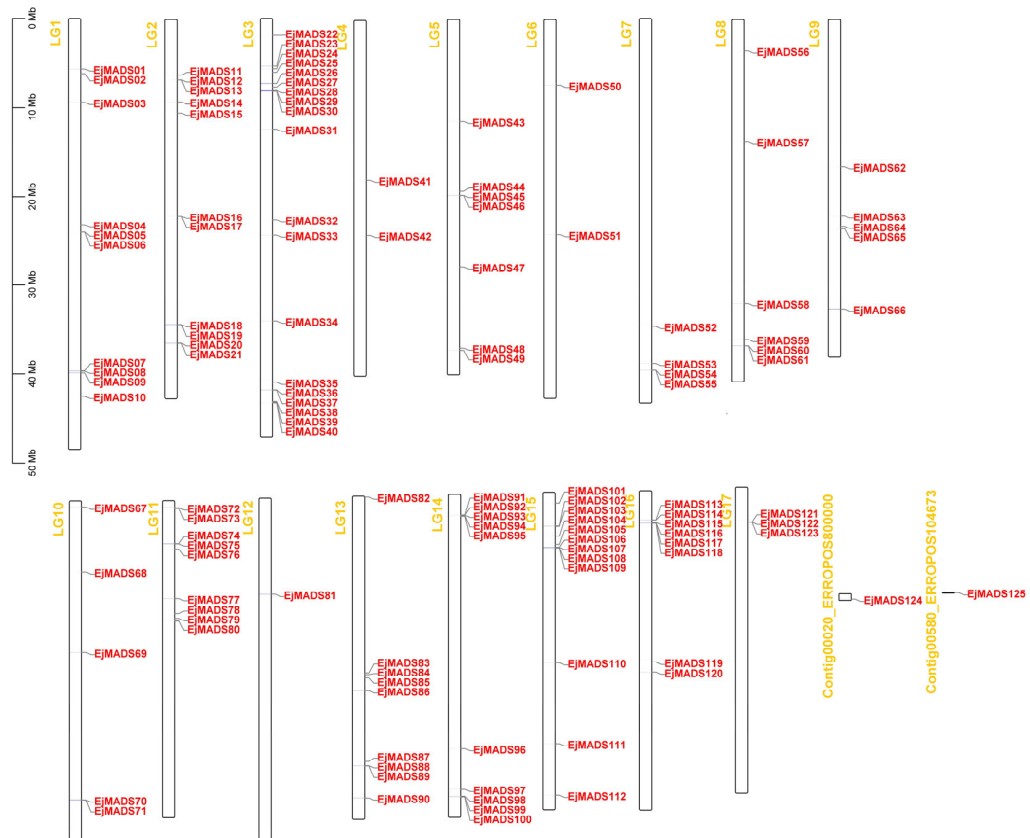

**Figure 3.** The chromosomal locations of *EjMADSs*. The chromosome numbers are equivalent to LG numbers. Red font represents genes, and yellow font represents chromosomes.

### 3.4. Gene Structures and Conserved Motifs

Structural information on 125 *EjMADSs* was obtained from the genome annotation file. The results showed that usually, the type II *EjMADSs* contain multiple introns, whereas the type I *EjMADSs* contain one or even no introns (Figure 4). By analyzing the protein sequences of 125 *EjMADSs* using the MEME online tool, combined with a phylogenetic tree of the *EjMADSs*, it was revealed that the closely related members had similar conserved motifs. Since type II EjMADS proteins possess both MADS and K domains, while type I proteins lack the K domain, all protein sequences except for EjMADS15, EjMADS32, EjMAD76, and EjMADS108, contained motif1, indicating that motif1 is the main component of the MADS domain, along with motif3, which collectively forms the MADS domain (Figure 4). Almost all type II EjMADS proteins contained motif2, motif5, and motif7, suggesting that these three motifs are important components of the K domain. Motif10 was unique in M$\alpha$ (Figure 4).

### 3.5. Gene Duplication and Synteny Analysis

The collinear relationships between the genes in the genomic context were analyzed to investigate gene expansion and duplication events. Among the *EjMADSs*, there were 69 collinear relationships involving 75 *EjMADSs* (Figure 5A). A tandem duplication event of 14 *EjMADSs* was also detected (Table S4). The collinear analysis showed that the expansion of the *EjMADSs* in the loquat genome was mostly caused by large-scale fragment duplication events, which may be related to the WGD event in loquat [32]. Most of the fragment duplication events originated from Chromosome 3. Only tandem duplication events occurred for chromosome 17, in which *EjMADS121*, *EjMADS122*, and *EjMADS123* were tandem duplicated in the M$\gamma$ subfamily (Figure 3). According to the collinearity analysis between species, it was observed that there was a higher degree of collinearity among the *MADS-box* genes when comparing the genomes of *Arabidopsis* and loquat, as

opposed to the comparison between the rice and loquat genomes, and this difference may be due to the closer evolutionary relationship between the dicotyledonous plants of loquat and *Arabidopsis* (Figure 5B). The genes with collinearity relationships may play similar functions in plant growth and development. To further assess the selection pressure on these genes that underwent duplication events, the Ka/Ks values were calculated for the repeated *EjMADSs*. The results revealed that all duplicated *EjMADS* gene pairs exhibited a ratio of <1, suggesting that the *EjMADS* genes were subjected to purifying selection and they were highly conserved in terms of their functions during evolution (Table S4).

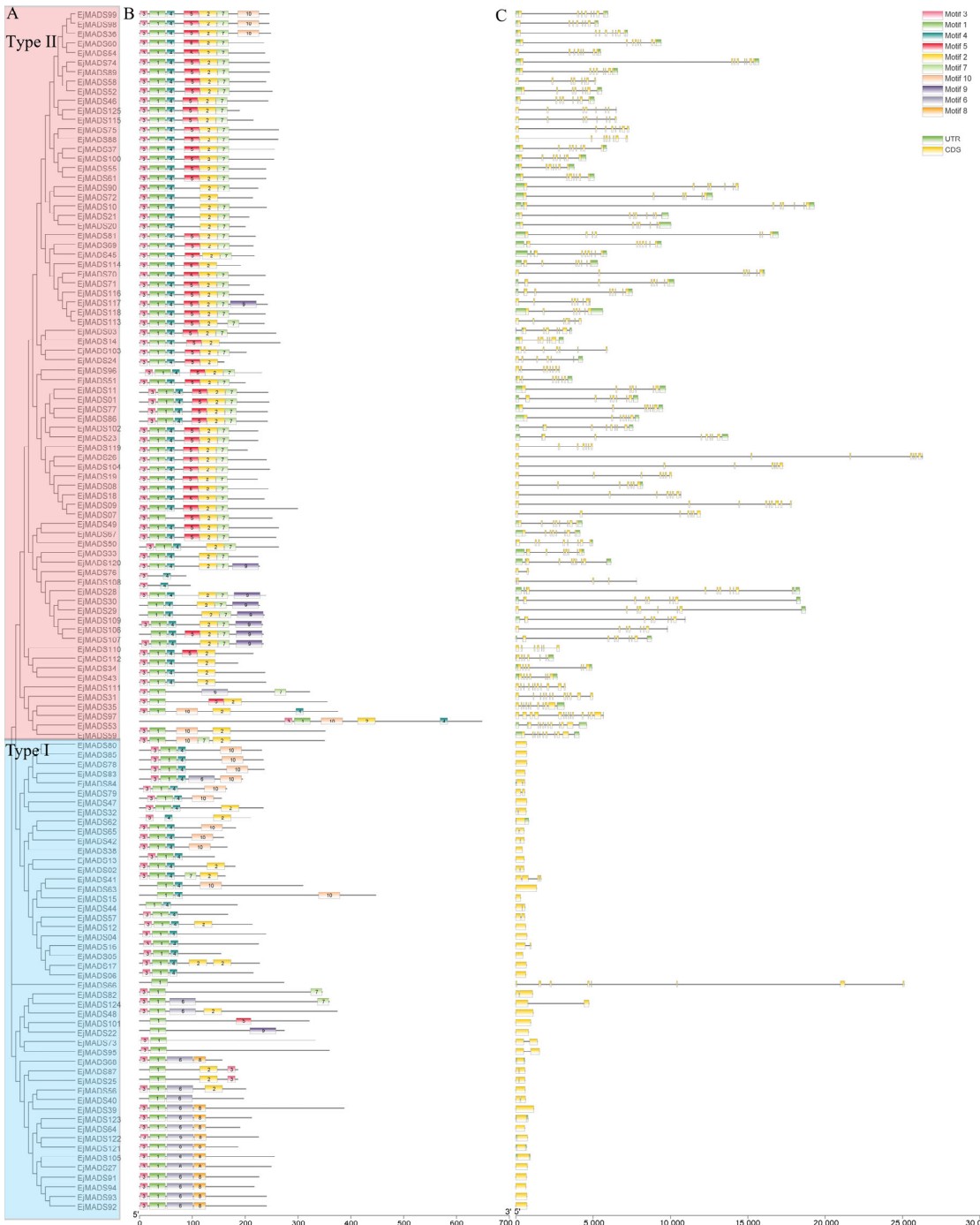

**Figure 4.** Conserved motifs and gene structures of the identified EjMADS proteins. (**A**) Phylogenetic relationships of EjMADS proteins. The blue rectangular overlay is type I, and the orange rectangular overlay is type II. (**B**) The colored boxes represent motifs. (**C**) Gene structures represented by exon–intron structures. Boxes represent exons, and lines represent introns.

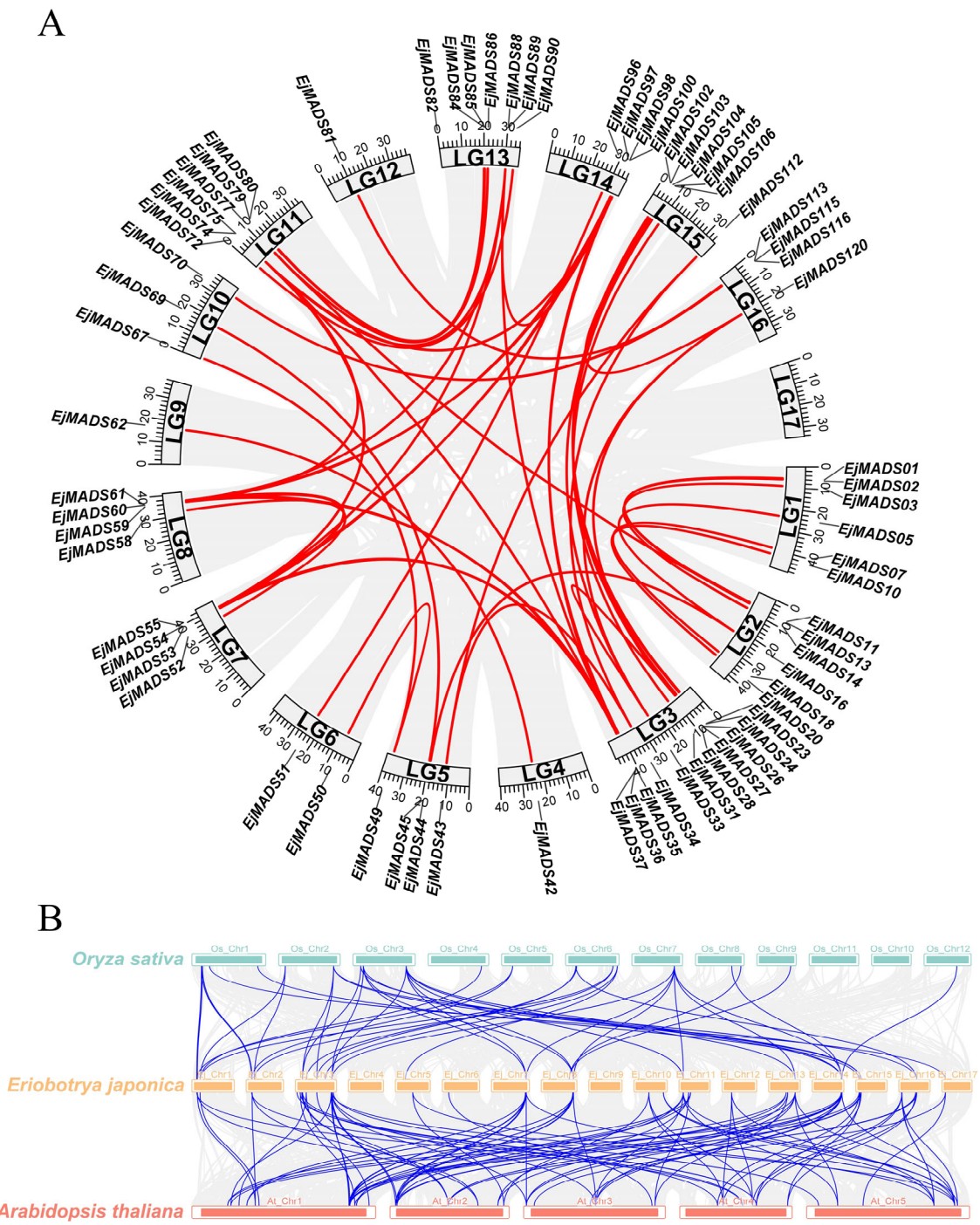

**Figure 5.** Collinearity of *MADS-box* genes. (**A**) Collinearity of *EjMADSs* in the loquat genome. The red line represents the collinearity between *EjMADSs*. The gray lines represent collinear relationships between other genes. (**B**) Collinearity of *MADS-box* genes among genomes of *Arabidopsis*, loquat, and rice. The blue line represents the collinearity between *MADS-box* members. Gray lines represent collinear relationships between other genes.

### 3.6. Expression Profiles in Different Tissues

To understand the potential functions of the *EjMADSs* in loquat, a re-analysis of previously published RNA-seq data was conducted. Specifically, this study aimed to analyze the expression patterns of *EjMADSs* across various organs and tissues, providing valuable insights into their potential roles in loquat development. Excluding some genes that were hardly expressed, 85 *EjMADSs* were expressed in at least one tis-

sue or an organ part (Figure 6). *EjMADSs* from the same subfamily appeared to be conservative in their expression patterns. Two AG-like genes (*EjMADS23/102*) and two TT16-like genes (*EjMADS14/3*) were highly expressed in seeds. Five *EjMADSs* (*EjMADS119/9/26/104/19*) of the ANR1 subfamily were highly expressed in roots. Five *EjMADSs* (*EjMADS59/31/97/35/53*) of the MIKC* subfamily were highly expressed in pollen. Three *EjMADSs* (*EjMADS81/118/116*) of the SOC1 subfamily had high expressions in young leaves. Four *EjMADSs* (*EjMADS60/98/54/89*) of the SEP subfamily had high expressions in green fruits. The *EjMADSs* belonging to the same subfamily likely contribute significantly to the development of the aforementioned tissues (Figure 6A). For another, the greatly expanded *EjMADSs* of the SVP subfamily seemed to be functionally differentiated, and its members are highly expressed in various tissues, including tissues of inflorescence, roots, stems, mature leaves, young leaves, and other tissue parts (Figure 6A).

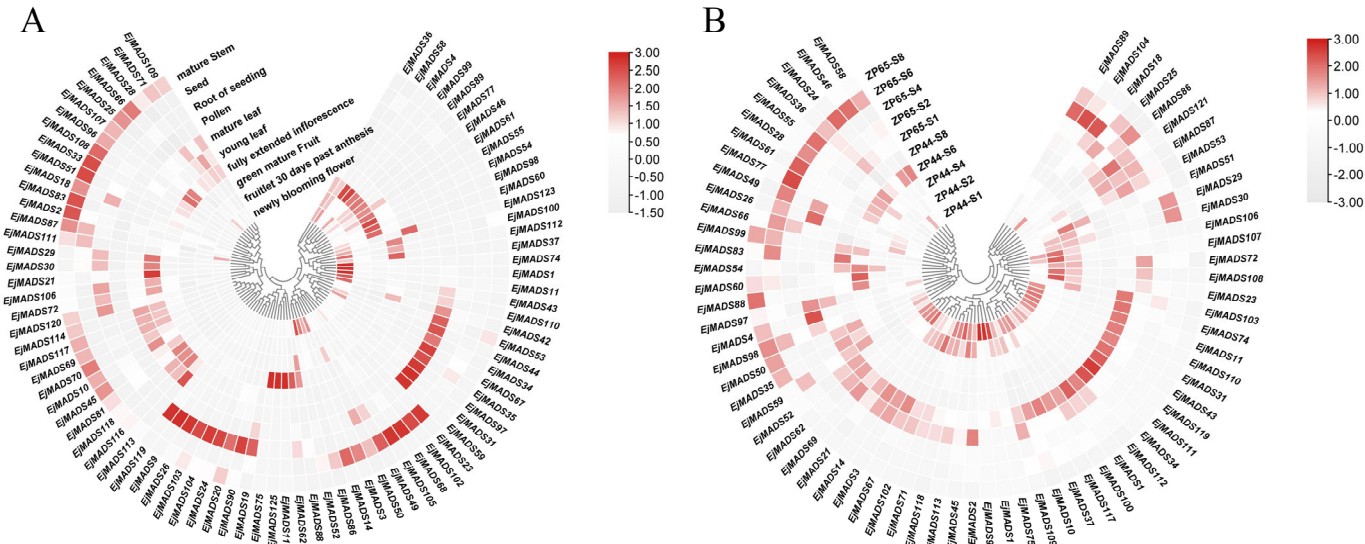

**Figure 6.** Expression profiles of *EjMADSs* in different loquat tissues. (**A**) Expression profiles of *EjMADSs* in stem, seed, root, leaf, pollen, flower, and fruit of loquat. (**B**) Expression profiles of *EjMADSs* at different development stages of two sister lines with large fruits and small fruits, respectively.

Fruit size is an important breeding target for horticultural plants, and fruit materials with extreme sizes are well research objects. Based on the transcriptome data of two sister lines, with one big-fruited and one small-fruited, at five different developmental stages, most *EjMADSs* showed similar expression patterns between big-fruited and small-fruited lines. However, the expression patterns of several *EjMADSs* (*EjMADS89/104/18/25/86/121/87/53*) showed differences between the two lines (Figure 6B). At the fruit expansion stage, the S6 stage reported previously [31], we observed that the expression levels of *EjMADSs* (*EjMADS46/24/49/67/55/61/77/86*) were found to be significantly higher in larger fruits (ZP65) compared to smaller fruits (ZP44). Consequently, they may be associated with the expansion process of loquat fruits (Figure 6B).

### 3.7. Expression Patterns in Flower Buds at Different Stages and in Different Tissues

Gene functions are typically reflected by the gene expression patterns. In this study, the tissue structures inside the bud were observed through a paraffin section of the stem apex tissues. It was confirmed that the flower bud differentiation began in mid-July, and obvious inflorescence became visible in mid-late September (Figure 7). To identify the type II members that potentially participate in flower bud differentiation or flower organ formation, their expressions were analyzed in various tissues, including the stem apical tissues at five developmental stages, and flower organ tissues, including petals, sepals, stamens, and pistils.

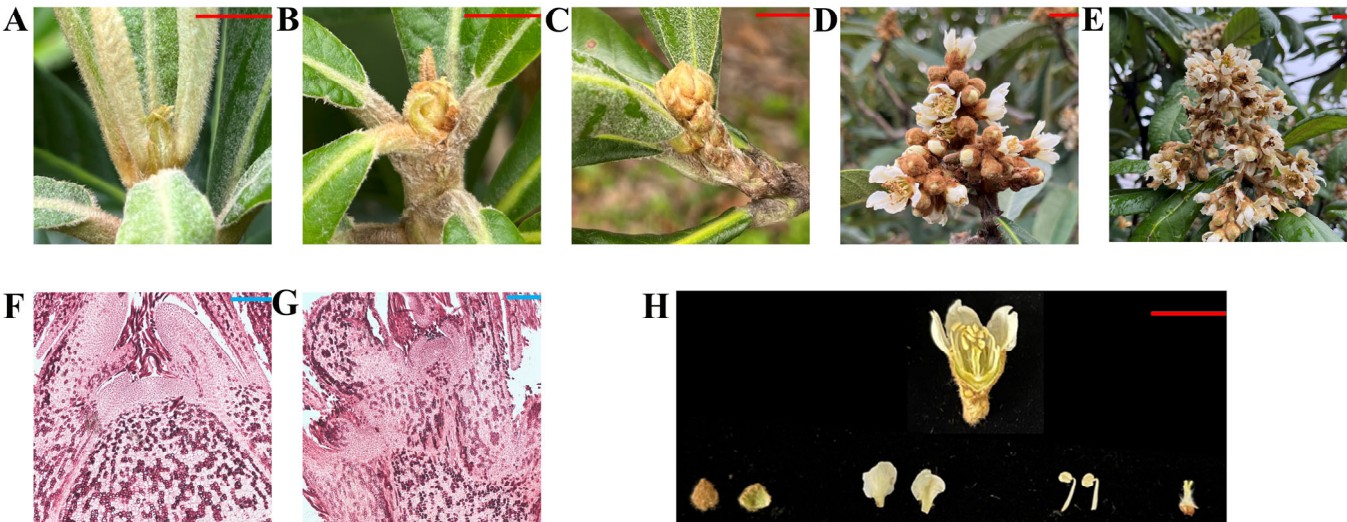

**Figure 7.** Stem apex tissue at different stages and flower organ tissue of loquat. (**A**) Vegetative bud. (**B**) Flower bud differentiation begins. (**C**) Visible flower bud. (**D**) The first flowering stage (the period when about 25% of the flowers of the whole tree are open). (**E**) Full-bloom stage (a period when about 75% of the flowers of the whole tree are open). (**F**) Paraffin section of vegetative bud. (**G**) Paraffin section of visible flower bud. (**H**) Flower organ tissue of loquat. The four flower organ tissues are sepals, petals, stamens, and pistils in the picture. The red bar represents 1 cm. The blue bar represents 200 μm.

In order to identify candidate *EjMADSs* involved in regulating the flower bud differentiation in loquat, nine genes were selected and their relative expression levels were analyzed at various stages of flower development. *EjAP1s* (*EjMADS55*, *EjMADS61*) were used as the mark of the onset of differentiation of flower buds. Based on paraffin sections and expressions of *EjAP1s*, stage 2 can be identified as the beginning of loquat flower bud differentiation (Figure 8). When the flower bud differentiation was underway, a notable increase in the expression levels of *EjMADS107* and *EjMADS109* was observed, indicating their potential significance in the regulation of this process in loquat (Figure 8). At stage 2, a significant upregulation in the expression of *EjMADS75* and *EjMADS106* was observed, while at stage 4, *EjMADS30* and *EjMADS89* exhibited high expression levels, suggesting their potential involvement in the developmental processes of loquat inflorescence (Figure 8).

To further explore the involvement of the *MADS-box* genes in the developmental processes of loquat flower organs, a total of 11 genes were analyzed, including two homologs of *AP1*, two homologs of *AP3* and *PI*, two homologs of *AG*, one homolog of *STK*, and four homologs of *SEP*. The results showed the two class A genes (*EjMADS55* and *EjMADS61*) varied significantly in their expression levels in sepals that were significantly higher than those in the other three tissue parts. *EjAP3* (*EjMADS43*), a class B gene, showed higher expression in petals and stamens, while *EjPI* (*MADS110*) showed the highest expression in petals. The two class C genes (*EjMADS1*, *EjMADS11*) showed higher expressions in both the stamen and pistil. The expression of the class D gene (*EjMADS23*) exhibited a significant increase specifically in the pistil when compared to the other three tissue parts. Four class E genes (*EjMADS58*, *EjMADS89*, *EjMADS52*, *EjMADS74*) were expressed in the sepals, petals, stamens, and pistils of loquat, suggesting that they may coordinate with other types of flower organ-characteristic genes to regulate the development of flower organs.

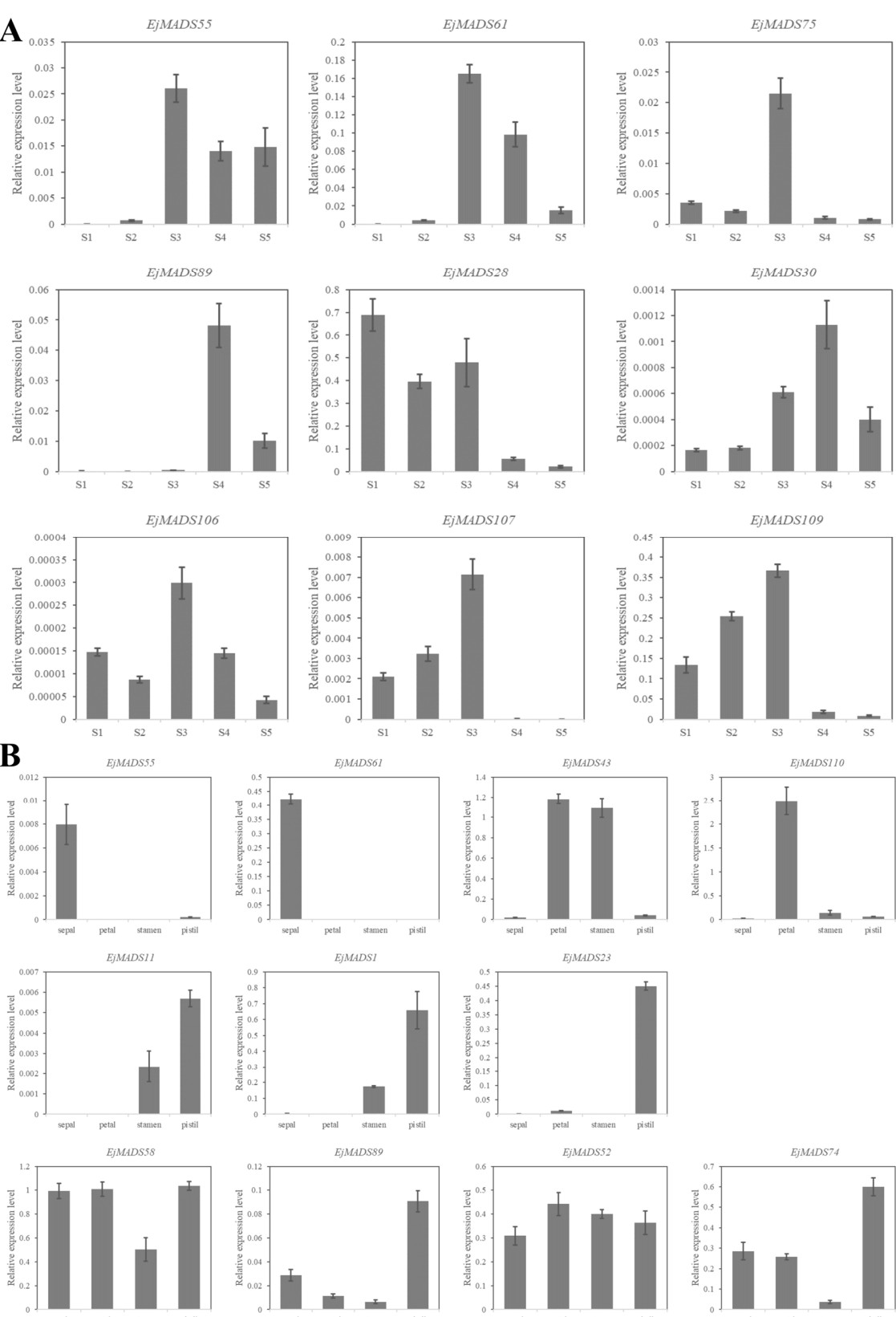

**Figure 8.** Expression analysis of *EjMADSs* in buds at different stages and in floral tissue parts. (**A**) Expression analysis of *EjMADSs* in buds at different stages using qRT-PCR. S1: vegetative bud; S2: flower bud differentiation begins; S3 visible flower bud; S4: the early flowering stage; S5: full-bloom stage. (**B**) qRT-PCR expression analysis of *EjMADSs* in floral tissue parts, including sepals, petals, stamens, and pistils. The error bars were standard errors.

## 4. Discussion

During long-term evolution and natural selection, eukaryotes have developed complex mechanisms for regulating gene expressions to adapt to changing living environments. The MADS-box proteins were widely known for their involvement in plant biological processes. At present, they have been identified in the genomes of many plant species, such as *Arabidopsis* [21], apple [40], pear [41], rice [42]. In loquat, the systematic characterization and analysis of the *MADS-box* genes, despite their significance, have not yet been documented. Our current study filled this gap by investigating the classification, features, phylogenetic relationships, evolution, as well as expressions of the *MADS-box* members in loquat. The results from this study are anticipated to contribute to an enhanced comprehension of this important gene family in loquat and present potential candidates implicated in the flower and fruit development in loquat.

### 4.1. Features of MADS-box Genes

In the current study, it was found that the type II genes identified in loquat, apple, and pear in Maloideae all greatly expanded compared with that in other Rosaceae species without experiencing a WGD event. The expanded *MADS-box* genes in loquat may be functionally differentiated, such as *EjSVP-1* (*EjMADS33*) and *EjSVP-2* (*EjMADS120*). In flowers, *EjSVP-1* is only expressed in the receptacle, whereas *EjSVP-2* seems to be expressed in all tissue parts including the receptacle, petal, stamen, and pistil [14], while some expanded *MADS-box* genes in loquat may have functional redundancy, such as *EjSOC1-1* (*EjMADS45*) and *EjSOC1-2* (*EjMADS69*), or *EjAP1-1* (*EjMADS55*) and *EjAP1-2* (*EjMADS61*), which play similar roles in flower bud differentiation [13,17].

For most plants such as lettuce, longan, pear, etc., the majority of type I genes do not possess an intron, whereas type II genes typically contain multiple introns [41,43,44]. This is similar in loquat, since only a few type I genes in loquat have introns, such as *EjMADS65*, *EjMADS66*, and *EjMADS124*, while *EjMADS66* has the most introns. The presence of introns is crucial for the occurrence of alternative splicing, which suggests that there may be different transcripts of the type II genes possibly with different functions [45].

### 4.2. Expression of EjMADSs in Different Tissues of Loquat

Currently, several important transcription factor families in plants have been extensively studied, such as MADS-box, R2R3-MYB, and WRKY, and they are involved in most of the biological processes in plants [4,46,47]. Figure 6A demonstrates that *EjMADSs* are expressed in many tissues, and the genes within the same subgroup tend to exhibit similar expressions, which is consistent with the result from a study in pineapple [48].

In horticultural crops, the attribute of fruit size captures significant attention and interest. In tomatoes, *LeMADS-RIN*, a *SEP1-like* gene, has been reported to possess a crucial function in the process of fruit development [49]. In this study, some *EjMADSs* exhibit distinct expression patterns throughout the progression of fruit development when comparing an extremely big-fruited line (ZP65) with a small-fruited line (ZP44), including that of a *SEP1-like* gene (*EjMADS89*) (Figure 6B). Compared with those in ZP44, *EjMADS89* was expressed at higher levels at the S4 and S6 stages in ZP65, implying that *EjMADS89* might have a crucial involvement in the development process of loquat fruits. A series of genes, including the *AGL6* homolog (*EjMADS46*), *AGL12* homolog (*EjMADS24*), *AGL6* homolog (*EjMADS49* and *EjMADS67*), *AP1* homolog (*EjMADS55* and *EjMADS61*), and *SHP1/2* homolog (*EjMADS77* and *EjMADS86*), showed higher expressions at the S6 stage (rapid fruit expansion stage) in ZP65 (Figures 6B and S1A), indicating a potential role in the expansion process of loquat fruits, and these genes deserve attention in the future breeding of loquat for its fruit size. In a study on Japanese pear (*Pyrus pyrifolia*), it was also found that *AP-like*, *AGL6*, and *SHP-like MADS-box* genes were highly expressed during the fruit growth and enlargement process [50]. On the other hand, *TAGL1* in tomatoes, as a homolog of *Arabidopsis SHP*, was found to promote fruit enlargement through transgenic *TAGL1* RNAi lines when compared to the wild type [51].

Recently, the integration of multiple omics data in loquat has shown that *EjTRN1* has great potential in promoting fruit enlargement [31]. Additionally, experimental evidence through Y1H (yeast one-hybrid) and VIGS (virus-induced gene silencing) has demonstrated that *EjBZR1* inhibits fruit enlargement by regulating *EjCYP90A* [37]. It is worth further investigating whether these *MADS-box* genes highly expressed during the rapid fruit enlargement stage in ZP65 are involved in regulating the genes related to fruit enlargement, such as *EjTRN1*, *EjBZR1*, *EjCYP90A*, and other genes.

*4.3. Functional Conservation of the ABCDE Model Genes in Loquat*

The studies of gene functions in model plants provide a valuable reference for the homologs in other plant species. The expression profiles of *EjMADSs* were investigated to understand their roles in flower organ development. From the expression patterns of the *EjMADSs* in these tissues, the flower organ identities were mostly conserved in loquat. However, some discrepancies were identified. In *Arabidopsis*, the class A genes were involved in the development of sepals and petals [52], but in loquat, there were two class A gene homologs *EjAP1s* (*EjMADS55* and *EjMADS61*) showing high expressions only in sepals, which may be caused by the different regulatory mechanisms among different species. On the other hand, the functions of the two *EjAP1s* as floral organ identity genes remain conserved (Figure 8A). In *Arabidopsis*, *AP1*, *CAL*, and *FUL* together specify floral meristem development [53,54]. *FT*, *LFY*, and *SOC1* can integrate signals from the environment to directly regulate *AP1*, thus precisely controlling flowering [6]. The MADS-box proteins can form tetrameric complexes, known as floral quartets, which determine the identity of floral organs [11]. In the ABCDE model, A + B determines petal formation, and B + C regulates stamen development [55]. A study in *Arabidopsis* indicated that the absence of B gene expression will lead to the loss of petals and stamens or the conversion of these floral organs into sepals and pistils, thus affecting the flower morphology and structure [10]. *EjAP3* (*EjMADS43*), a class B gene homolog in loquat, exhibited high expression levels in both the petals and stamens, demonstrating that it participates in petal and stamen identities in loquat. However, the class B gene homolog *EjPI* (*EjMADS110*) seems to have undergone functional changes in loquat, since it was only highly expressed in petals, whereas *PI* can regulate both petals and stamen identities in *Arabidopsis* [56]. In model plants, *AP1* and *LFY* redundantly activate *AP3* and *PI* to promote the formation of floral organs [57,58]. In a recent study, it was found that *AP1* and *LFY* can also reduce the inhibition of *ZP1* and *ZFP8* on *AP3*, *PI*, and *AG* by suppressing their expressions in the floral meristem [59]. In loquat, the class C homologs *EjAGs* (*EjMADS1* and *EjMADS11*) exhibited high expression levels in both stamens and pistils, the class D homolog *EjSTK* (*EjMADS23*) was highly expressed only in pistils, while these class C/D genes were not expressed or had low expressions in sepals and petals. This observation aligns with findings from a previous study in *Antirrhinum*, where it was demonstrated that the class C/D genes primarily control stamen and pistil development [60]. In *Arabidopsis*, E genes (*SEP1-4*) are involved in the development of every flower organ, and they are functionally redundant [61]. Additionally, B and C class genes are synergistically activated by *SEP3* and *LFY* [62]. In loquat flowers, five class E homologs, including *EjMADS58*, *EjMADS89*, *EjMADS52*, and *EjMADS74*, were identified in loquat, and they may function as cofactors. By analyzing the expression patterns of these genes, we can better understand the morphogenetic mechanisms of loquat flower, providing a theoretic foundation for the improvement of flower-related traits.

Specific environmental conditions provide plants with signals to initiate flowering. Compared to other Rosaceae fruit trees such as apple and pear, loquat has a unique flowering time, occurring in autumn [63,64]. The formation of floral organs is a prerequisite for flowering. Investigating the impact of different environmental conditions, such as temperature and light duration, on the floral organ genes, as well as delving into the molecular regulatory networks associated with floral organ genes discovered in model

plants, may help elucidate the molecular mechanisms underlying the unique flowering phenomenon in loquats.

## 5. Conclusions

Collectively, we conducted a thorough examination of the *MADS-box* genes in loquat, covering their family characteristics, syntenic relationships, and expression patterns. Based on the transcriptome data, *EjMADS46*, *EjMADS24*, *EjMADS49*, *EjMADS67*, *EjMADS55*, *EjMADS61*, *EjMADS77*, and *EjMADS86* had higher expressions at the rapid fruit expansion stage in the big-fruited line, indicating their role in the expansion process of loquat fruits. These genes may deserve to be prioritized in future breeding for fruit size. Based on the qRT-PCR experiments exploring the expression patterns of *EjMADSs* in floral tissue parts and in buds at different stages, it was found that *EjMADS107* and *EjMADS109* may potentially assume a crucial function in flower bud differentiation, and that the ABCDE model homologous genes in loquat are conserved in their expression patterns of loquat floral tissue parts. Results from this study are expected to contribute to a more comprehensive comprehension of the involvement of the *MADS-box* genes in the development of flowers and fruits in loquat.

**Supplementary Materials:** The following supporting information can be downloaded at https://www.mdpi.com/article/10.3390/agronomy13112709/s1: Figure S1: Phylogenetic relationships of MADS-box from Roseaceae species and *Arabidopsis thaliana*, Table S1: List of primers used for qRT-PCR in this study, Table S2: MADS-box proteins from all Rosaceae species in this study, Table S3: Basic information about the *EjMADSs* in loquat, Table S4: Positive selection analysis of the *MADS-box* gene family in loquat.

**Author Contributions:** Conceptualization and writing—review and editing, Z.P.; methodology, formal analysis, software, and writing—original draft preparation, W.L. and X.L.; supervision and validation, Z.P. and X.Y.; investigation, W.W., Y.J., W.S. and C.Z.; resources, S.L., Z.P. and X.Y.; funding acquisition, Z.P. and X.Y. All authors have read and agreed to the published version of the manuscript.

**Funding:** This research was funded by the Key-Area Research and Development Program of Guangdong Province (2022B0202070002), National Natural Science Foundation of China (32202429), Natural Science Foundation of Guangdong Province (2022A1515012273), Guangzhou Science and Technology Project (202201010345), the Open Fund of the Guangdong Provincial Key Laboratory of Utilization and Conservation of Food and Medicinal Resources in Northern Region (Grant number FMR2022009Z).

**Data Availability Statement:** Not applicable.

**Conflicts of Interest:** The authors declare no conflict of interest.

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
