# Peer review of "Genome-Wide Characterization of MADS-box Genes Identifies Candidates Associated with Flower and Fruit Development in Loquat (Eriobotrya japonica Lindl.)"

_agronomy, doi:10.3390/agronomy13112709_

Round 1

Reviewer 1 Report

Comments and Suggestions for Authors

This manuscript entitled "Genome-wide characterization of MASD-box genes identified candidates associated with flower and fruit development in loquat." provides a genome wide analysis for identify and characterize the MADS-box gene family in loquat, which is beneficial for development of flower in loquat.

I think this topic is original and relevant in this field. To date, there are few studies about the MADS-box gene family in loquat, this study focused on MADS-box gene family.

In general, The manuscript was written well, However, there were still several small mistakes.

1. Line 93: the fresh samples were frozen by liquid nitrogen, then kept at -80? please provide more details.

2. Line 145: add reference about ΔΔCT method. 

Analysis of relative gene expression data using real-time quantitative PCR and the 2− ΔΔCT method 2001, Methods.

3. line 337, what is the word "he" mean?  the word is "it"?

Reviewer 2 Report

Comments and Suggestions for Authors

In this work the authors identified MADS-box genes in loquat. Generally, the they described the paper very well. Mostly, there was no missing point. However, related to title of the work, I prefer to see some more qPCR from fruit development tissues.  

The title of this work is,  Genome-wide characterization of MADS-box genes identifies candidates associated with flower and fruit development in loquat.

But, especially in qRT-PCR analysis, I could not see any qPCR- analysis from fruit development process. I think, the MS will be much stronger if the authors include a couple of more genes from green mature fruit and 30 days post anthesis fruit tissues.

Moreover, do you have any samples from the plant with different fruit sizes? IT might be also useful to add a couple of gene expression related with fruit size. But, if you don’t have sample, no problem.

Reviewer 3 Report

Comments and Suggestions for Authors

This study focuses on the examination of MADS-box transcription factors and their involvement in the process of flower organogenesis in loquat. The primary objectives are to identify and characterise the MADS-box gene family in loquat and to gain insights into their potential roles in flower and fruit development. A total of 125 EjMADS-box genes have been identified and categorised into two groups: type I and type II members. It is noteworthy that the type II MADS-box genes have undergone expansion, likely a consequence of whole-genome duplication within the Maloideae lineage. The analysis of gene expression has not only reaffirmed the presence of ABCDE model homologs in loquat but has also pinpointed candidate genes that play pivotal roles in flower bud differentiation and fruit expansion. The research serves as a comprehensive analysis of MADS-box genes in loquat, thus bridging an existing knowledge gap.

While this study offers a well-defined research objective, there are some minor points to consider:

1.         Although the study identifies candidate genes, it refrains from providing experimental validation of their functions. As such, supplementary experiments are essential to confirm their roles in flower and fruit development. Alternatively, a thorough discussion based on past research findings could compensate for this gap.

2.         This work establishes a foundational understanding of the MADS-box gene family in loquat but does not delve deeply into the intricate molecular mechanisms underpinning their functions. A more extensive discussion and suggestions for future research avenues may serve to elucidate these mechanisms.

3.         The manuscript alludes to a genome-wide analysis conducted to identify and characterise MADS-box genes in loquat, yet it would be beneficial to explicitly state the specific research objectives or questions the study aimed to address. This clarity at the outset would facilitate readers' comprehension of the study's purpose.

4.         Although the study acknowledges the importance of MADS-box genes in flower development, it neglects to delve into the intricate gene regulatory networks and interactions at play. This omission hampers a comprehensive understanding of these processes. Including detailed discussions based on prior studies could be a remedy.

5.         Redundancy is a minor issue in the manuscript. For example, the details about the expression patterns of EjMADSs in different tissues and their potential roles in loquat development are reiterated in multiple sections. To enhance conciseness, this repetition could be condensed.

6.         The English writing in the manuscript is clear and well-structured. However, there is room for improvement in simplifying complex terms and enhancing reader engagement. Furthermore, maintaining consistent verb tenses is important. These refinements would enhance the accessibility of the study to a broader audience.

Comments on the Quality of English Language

 The English writing in the manuscript is clear and well-structured. However, there is room for improvement in simplifying complex terms and enhancing reader engagement. Furthermore, maintaining consistent verb tenses is important. These refinements would enhance the accessibility of the study to a broader audience.

Reviewer 4 Report

Comments and Suggestions for Authors

Dear Prof. Dr. Editor of Agronomy Journal,

I am writing you regarding Manuscript ID: agronomy-2668734 entitled: Genome-wide characterization of MADS-box genes identifies candidates associated with flower and fruit development in loquat (Eriobotrya japonica Lindl.)" which was submitted to the Agronomy Journal.

In this manuscript, the author's analysis of Genome-wide MADS-box genes identifies candidates associated with flower and fruit development in loquat (Eriobotrya japonica Lindl.). The manuscript is suitable for publication in Agronomy.

I have gone through this work. My decision is accepted with minor revisions for this work. The reason for that is as follows:

The manuscript deals with the

First: Title: need change to:

Genome-wide characterization of MADS-box genes associated with flower and fruit development in loquat (Eriobotrya japonica Lindl.) 

Second Abstract, Keywords and Introduction:

2) has some minor corrections as in the attached file.

Third: The objectives of the study

3) Need to rewrite in clear points. 

Fourth: Materials and Methods

4) has some minor corrections as in the attached file. 

Results and discussion

5)  has some minor corrections as in the attached file.

References

6) It has some minor corrections as in the attached file and please check J guidelines.

Thank you for suggesting me as a reviewer for this paper.

with best regards

Comments on the Quality of English Language

minor revision
